# Prediction of algal blooms via data-driven machine learning models: An evaluation using data from a well monitored mesotrophic lake

Shuqi Lin[1,3*], Donald C. Pierson[1], Jorrit P. Mesman [1,2]

[1]Erken Laboratory and Limnology Department, Uppsala University, Uppsala, Sweden

[2]Département F.-A. Forel des sciences de l'environnement et de l'eau, Université de Genève, Genève, Switzerland

[3]Environment and Climate Change Canada, Canada Centre for Inland Waters, Burlington, ON, Canada, L7R 4A6

*Correspondence to*: Shuqi Lin (Shuqi.Lin@ec.gc.ca)

**Abstract.** With the increasing lake monitoring data, data-driven machine learning (ML) models might be able to capture the complex algal bloom dynamics that cannot be completely described in process-based (PB) models. We applied two ML models, Gradient Boost Regressor (GBR) and Long Short-Term Memory (LSTM) network, to predict algal blooms and seasonal changes in algal chlorophyll concentrations (*Chl*) in a mesotrophic lake. Three predictive workflows were tested, one based solely on available measurements, and the others applying a two-step approach, first estimating lake nutrients that have limited observations, and then predicting *Chl* using observed and pre-generated environmental factors. The third workflow was developed by using hydrodynamic data derived from a PB model as additional training features in the two-step ML approach. The performance of the ML models was superior to a PB model in predicting nutrients and *Chl*. The hybrid model further improved the prediction of the timing and magnitude of algal blooms. A data sparsity test based on shuffling the order of training and testing years showed the accuracy of ML models decreased with increasing sample interval, and model performance varied with training/testing year combinations.

## 1 Introduction

Harmful algal blooms, which are a serious threat to natural water systems, have been increasing throughout the world (Burford et al., 2020; Watson et al., 2016), primarily as a consequence of both climate change and increased nutrient loading from anthropogenic activities (Brookes and Carey, 2011; Paerl and Huisman, 2008). Moreover, as indicated by Carey et al. (2012) and Huisman et al. (2018), more intense and longer periods of thermal stratification could potentially specifically favour blooms of toxic cyanobacteria. To better manage and mitigate the effects of algal

blooms, methods to forecast their timing and magnitude are needed. However, the factors regulating algal blooms are
complex, variable and site-specific, often involving high-order interactions of environmental factors and
biogeochemical processes (Reichwaldt and Ghadouani, 2012; Richardson et al., 2018).
Process Based (PB) models encode our understanding of biogeochemical processes into a framework of numerical
formulations, but these are inevitable simplifications that lead to an incomplete description of complex
biogeochemical interactions and low level of model confidence (Elliott, 2012). Based on innovative data mining and
statistical techniques, data-driven machine learning (ML) models have been applied to identify patterns within
observed data (Peretyatko et al., 2012; Mellios et al., 2020), and with the recent proliferation of lake monitoring data
(Marcé et al., 2016), ML models have been applied, as an alternative to PB models for bloom prediction (Rousso et
al., 2020). Previously applied ML models, including Random Forest (Nelson et al., 2018), Support Vector Machine
(Jimeno-Sáez et al., 2020), and Artificial Neural Network (Xiao et al., 2017; Recknagel et al., 1998; Wei et al., 2001),
can improve predictions of the timing and seasonality of algal *Chl* pattern, apparently by accounting for complexity
that is difficult to encode within the framework of a PB model. However, a downside of data-driven ML models is
that they lack the interpretability and generalization found in the explicit structure of the PB model. In recent years,
process-guided-deep learning (PGDL) model emerged and was applied to water temperature (Jia et al., 2019; Read et
al., 2019) and water quality (Hanson et al., 2020) simulations, which explicitly combine well-defined physical theories
into the training of ML models, enhancing their interpretability. While this approach has achieved promising results,
it is difficult to apply it to phytoplankton dynamics due to numerous nonlinear interactions within the biogeochemical
cycles and the difficulty in defining a measurable processes or mass balances that can be used as a physical constraint
on knowledge-guided decisions. Also, the sparsity of lake water quality (e.g., nutrients, *Chl* concentration)
observations can limit the application of ML models in algal bloom modelling (Rousso et al., 2020).
In this study, our objectives are to (1) apply the ML models to predict algal bloom in a well-monitored mesotrophic
lake; (2) evaluate model performance and assess model uncertainties; (3) explore the approaches to improve the model
performance and widen the model applications. We first tested the ability of ML models in predicting algal *Chl*
concentrations via available environmental factors, including observed lake nutrients data, and then proposed a two-
step ML approach for predicting algal dynamics that: first estimates lake nutrient concentrations which often have
limited observations and secondly predicts variations in algal *Chl* using these pre-generated nutrient concentrations
combined with other observed environmental factors that are collected at higher frequency. We also tested a simple
hybrid model architecture that by adding hydrodynamic features derived from the PB model into the training features
of the two-step ML approach, allowing us to include additional information describing physical lake processes
expected to affect variations in algal growth and succession in the machine learning prediction.
We applied the above workflows to predict changing *Chl* concentration, as a proxy for the occurrence of algal blooms,
via Gradient Boost Regressor (GBR) and Long Short-term Memory network (LSTM). Two shuffling year tests were
conducted. One assessed the uncertainty of ML models in predicting *Chl* during the same two-year period and the
other evaluated the sensitivity of ML accuracy to various training/testing year combinations and lake nutrient sampling
intervals. Model performance and potential applications in algal bloom forecasting are discussed.
**2 Methods**
**2.1 Study site**
The study site, Lake Erken, is a mesotrophic lake located in east-central Sweden, that has a surface area of 24 km$^2$, a
maximum depth of 21 m and an average retention time of 7 years. The lake is dimictic with seasonal stratification
commonly beginning in May-June and ending in August-September. The onset of ice cover usually begins in
December-February and the loss of ice occurs in Mar-April (Persson and Jones, 2008). Located near the Baltic coast,
Lake Erken is wind exposed, and susceptible to periodic wind-induced turbulent mixing.
Changes in algal *Chl* in Lake Erken have a typical seasonal pattern, with spring and summer peaks in concentration
(Pettersson et al., 2003). Spring blooms are dominated by dinoflagellates and diatoms (Pettersson, 1985), and initiated
by overwinter species from the last autumn (Yang et al., 2016). Cyanobacteria dominate summer peaks in *Chl*, given
that they can optimize their vertical position in regarding to nutrients and light (Paerl, 1988; Pierson et al., 1992).

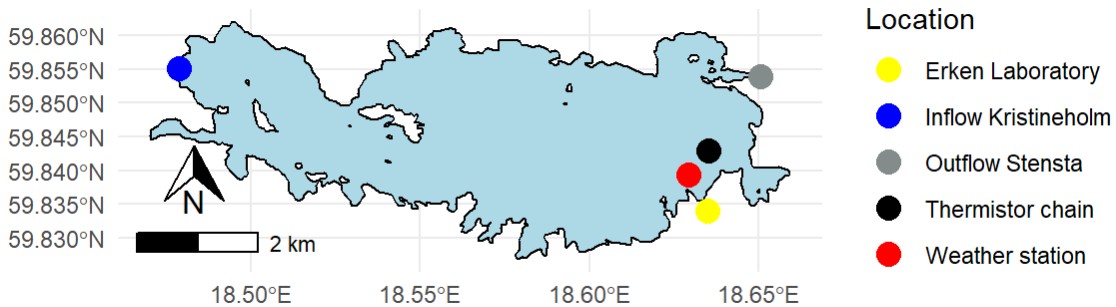


**Figure 1.** Map of Lake Erken. The locations of the monitoring systems are shown.
**2.2 Data**
Lake Erken has a long running automated monitoring program that provides hourly meteorological data, water
temperature profiles between 0.5 and 15 m at 0.5 m intervals and the flow from the inflow and outflow (Fig.1). A
manual sampling program collects samples during ice-free time at 5-7 days intervals for all major nutrient
concentrations (e.g., $NO_X$, $NH_4$, $PO_4$, Total P, Si, etc.), dissolved oxygen ($O_2$), and *Chl* concentration. The timing of
the onset and loss of ice cover are also monitored yearly by the lab. More detailed information on the sampling program
is in Supporting Information (See Text S1) and Moras et al. (2019).
**2.3 Modelling Methods**
2.3.1 Process-based (PB) lake model
In this study, a PB hydrodynamic lake model, GOTM (General Ocean Turbulence Model) (Burchard et al., 1999),
was used to generate water temperature profiles, and other hydrodynamic metrics. GOTM also served as the
foundation of water quality simulations made with the SELMAPROTBAS model (Mesman et al., 2022) that is coupled
to GOTM through the Framework for Aquatic Biogeochemical Models FABM (Bruggeman and Bolding, 2014).
2.3.2 Data-driven machine learning (ML) models
Tree models have been widely applied in modelling phytoplankton dynamics in freshwater systems (Harris and
Graham, 2017; Fornarelli et al., 2013; Rousso et al., 2020). Gradient Boosting Regressor (GBR) is one of these tree
models, iteratively generating an ensemble of estimator trees with each tree improving upon the performance of the
previous. The details about GBR model can be found in Friedman (2001). The hyperparameters in GBR are optimized
via *RandomizedSearchCV* function within Scikit-Learn library. The loss function of model is chosen as 'huber', which
is a combination of the squared error and absolute error of regression. Since the target variable in our research *Chl*
concentration has peak values during algal blooms which could be regarded as outliers, the 'huber' loss function is
more robust and gives greater weight to peak values than the mean squared error function.
Long short-term memory (LSTM) network is part of a class of deep learning architectures, called recurrent neural
network (RNN), built for sequential and timeseries modelling (Hochreiter and Schmidhuber, 1997). The core concepts
of LSTM are the cell and hidden states, and its three gates (input gate, forget gate, and output gate; See Fig. S2).
Essentially, the LSTM model defines a transition relationship for a hidden representation through a LSTM cell which
combines the input features at each time step with the inherited information from previous time steps. This architecture
is suitable for extracting information from sequential data (Rahmani et al., 2020; Read et al., 2019). The
hyperparameter settings in LSTM can be found in Supporting Information (See Text S2).
Compared to GBR model, LSTM has more complex model architectures, carrying the 'memory' from the previous
time steps. In this study, GBR and LSTM were applied, respectively, to assess the performance of ML models with
and without 'memory'. Both ML models are built in Python using the Scikit-Learn (https://scikit-learn.org/stable/, last
access: September, 2022) and TensorFlow (https://www.tensorflow.org/, last access: September, 2022) libraries.
**2.4 Design of predictive workflows and shuffling year data sparsity tests**
In this study, we tested three workflows using a dataset split for training (years 2004-2016) and testing (years 2017-
2020). In all three workflows, a 5-fold cross-validation using the training dataset was used to optimize the
hyperparameters in the ML models. Workflow 1 directly predicts *Chl* concentration based on available environmental
observations (Table 1). The training and testing datasets were limited by the frequency of lake nutrient observations
which resulted in 5-7 day gaps between data points. The time step of LSTM was set to 1, that is, the environmental
factors on the target date and previous observation date, which may be 5-7 days ago, were used to train the model and
make predictions.
In workflow 2 and 3, a two-step approach was applied (Table 1). Daily measurements of physical factors were used
to pre-generate daily variations in lake nutrients via separate ML models, and the ML models were trained at a daily
time step using the measured environmental factors and pre-generated nutrient concentrations. The time step of LSTM
was then set to 7 days.
In workflow 3, three hydrodynamic features, i.e., mixing layer depth ($z_e$), Wedderburn number ($W_n$), and the seasonal
thermocline depth (*thermD*), derived from the GOTM model were regarded as daily training features in the two-step
ML approach. The definitions and calculations of these features are explained in SI (2.5 Feature selection and
processing for ML models, Text S3)
Following the two-step approach and using workflow 3, we set up two tests. (1) To assess the uncertainty induced by
variations in the data used to train the ML models, we shuffled the training years, randomly taking 13 years out of
2004-2018 dataset 30 times, and tested the model predictions of *Chl* during 2019-2020. And, (2) to test if the workflow
could be used for other water systems which may have less frequent lake nutrient monitoring data, we conducted a
data sparsity test that evaluated the sensitivity of models to the lake nutrient and *Chl* sampling interval. For this test
the lake nutrient and *Chl* concentration observations in training dataset was down-sampled to a 7-day, 14-day, 21-

day, 28-day, and 35-day sampling interval. Then for each sampling interval using the 2004-2020 dataset, *Chl* was predicted for different consecutive 4-year periods when the ML models were trained by the remaining 13 years of data. Data shuffling was conducted 13 times so that every 4-year period in our dataset was tested.

**Table 1** List of training features and target variables in each workflow. Blue indicates training features, red indicates target variables, purple indicates the variables are the target variables in step 1 used to produce daily a training feature for use in step 2. The order of nutrient model sequence is from the top to bottom based on its position in the table (NOx to Si).

| variables | Sample interval | workflow 1 | workflow 2 Step 1 | workflow 2 Step 2 | workflow 3 Step 1 | workflow 3 Step 2 |
|---|---|---|---|---|---|---|
| Inflow | Daily | blue | blue | blue | blue | blue |
| Meteorological data (Air temperature, wind speed, shortwave radiation, precipitation, humidity, cloud cover) | Daily | blue | blue | blue | blue | blue |
| $\Delta T$ | Daily | blue | blue | blue | blue | blue |
| Ice duration | Daily | blue | blue | blue | blue | blue |
| Days from ice-off date | Daily | blue | blue | blue | blue | blue |
| $z_e$ | Daily | | | | blue | blue |
| $W_n$ | Daily | | | | blue | blue |
| *thermD* | Daily | | | | blue | blue |
| NOx | 1-2 weeks | blue | purple | blue | purple | blue |
| $O_2$ | 1-2 weeks | blue | purple | blue | purple | blue |
| $PO_4$ | 1-2 weeks | blue | purple | blue | purple | blue |
| Total P | 1-2 weeks | blue | purple | blue | purple | blue |
| $NH_4$ | 1-2 weeks | blue | purple | blue | purple | blue |
| Si | 1-2 weeks | blue | purple | blue | purple | blue |
| Chl | 1-2 weeks | red | | red | | red |

## 2.5 Feature selection and processing for ML models

The feature selection process is based on some a priori knowledge of the underlying phenomena related to algal blooms. All workflows made use of the daily automated monitoring data. In addition, the temperature difference ($\Delta T$) between surface water (averaged over the upper 3 m) and bottom water (15 m) was also used to represent the thermal structure of the lake., and the duration of ice cover in the previous winter, and the number of days from ice-off date were used.

In workflow 2 and 3 nutrients are predicted sequentially, with each pre-generated nutrient predictions included in the training data of the next nutrient prediction (Table 1). Workflow 3 added $z_e$, computed using the GOTM simulated vertical eddy diffusivity ($K_z$) profiles, *thermD*, estimated using Lake Analyzer (Read et al., 2011) based on GOTM simulated temperature profile, and $W_n$, a dimensionless parameter measuring the balance between wind stress and the pressure gradient resulting from the slope of the interface (See Text S3, SI), as additional daily training features.

**2.6 Evaluating metrics**
Model performance was evaluated by comparing the simulated and measured *Chl* concentrations, and by calculating
the mean absolute error (*MAE*), root means square error (*RMSE*), and correlation coefficient ($R^2$). To evaluate the
accuracy of the model in detecting the onset of an algal bloom, we calculated a confusion matrix in workflows 2 and
3, where the observations were linearly interpolated to daily values, and predicted daily *Chl* concentration were
smoothed with a 7-day rolling mean. Using these data, the onset of a bloom was categorized as occurring when the
daily change of *Chl* (*ΔChl*) exceeded a threshold, 0.35 mg m$^{-3}$ day$^{-1}$. This works well in Lake Erken where *Chl*
concentrations are frequently monitored (near weekly), and the linear interpolation can be expected to be reasonably
representative of the *Chl* concentrations between measured samples. Considering the randomization in the ML models,
we also add a 3-day window on the bloom onset prediction, that is, we considered the prediction of a bloom valid if
the measured data suggested a bloom the day before or after the simulated onset. We used the True Positive Rate
(TPR), False Positive Rate (FPR), and modified accuracy (Kappa) which considers the possibility of the agreement
occurring by chance (McHugh, 2012), to identify the potential of ML models to correctly capture the algal bloom
onset (See Table S1, SI). A model with 100% TPR, 0% FPR, and 100% Kappa would constitute a perfect fit.
**3 Results**
**3.1 Workflow 1: Direct prediction based on observations**
In workflow 1, both GBR and LSTM clearly reproduced spring and summer blooms (Fig. 2a) but underestimated the
intensity of blooms (Fig. 2a, b). Neither ML model captured the extraordinarily high *Chl* (~15-30 mg m$^{-3}$) in the
summer of 2019. Although the abnormal summer bloom in 2019 could contribute to the higher *RMSE* and *MAE* in the
testing dataset than the mean values in the training dataset, the cross-validation on the training dataset (See Table S2,
SI) shows what appears possibly to be overfitting issue in both models. The achieved accuracy of models is attributed
to the daily availability of physical inputs, and the fact that in Lake Erken water samples are collected frequently at 5-
7 days intervals. Workflow 1 may be most valuable in reconstructing previous variations in algal *Chl*, filling the gaps
between measured *Chl* observations and feature importance ranking (See Fig. S4, SI). But when using this workflow,
future forecasts will be limited by the absence of future nutrient data.

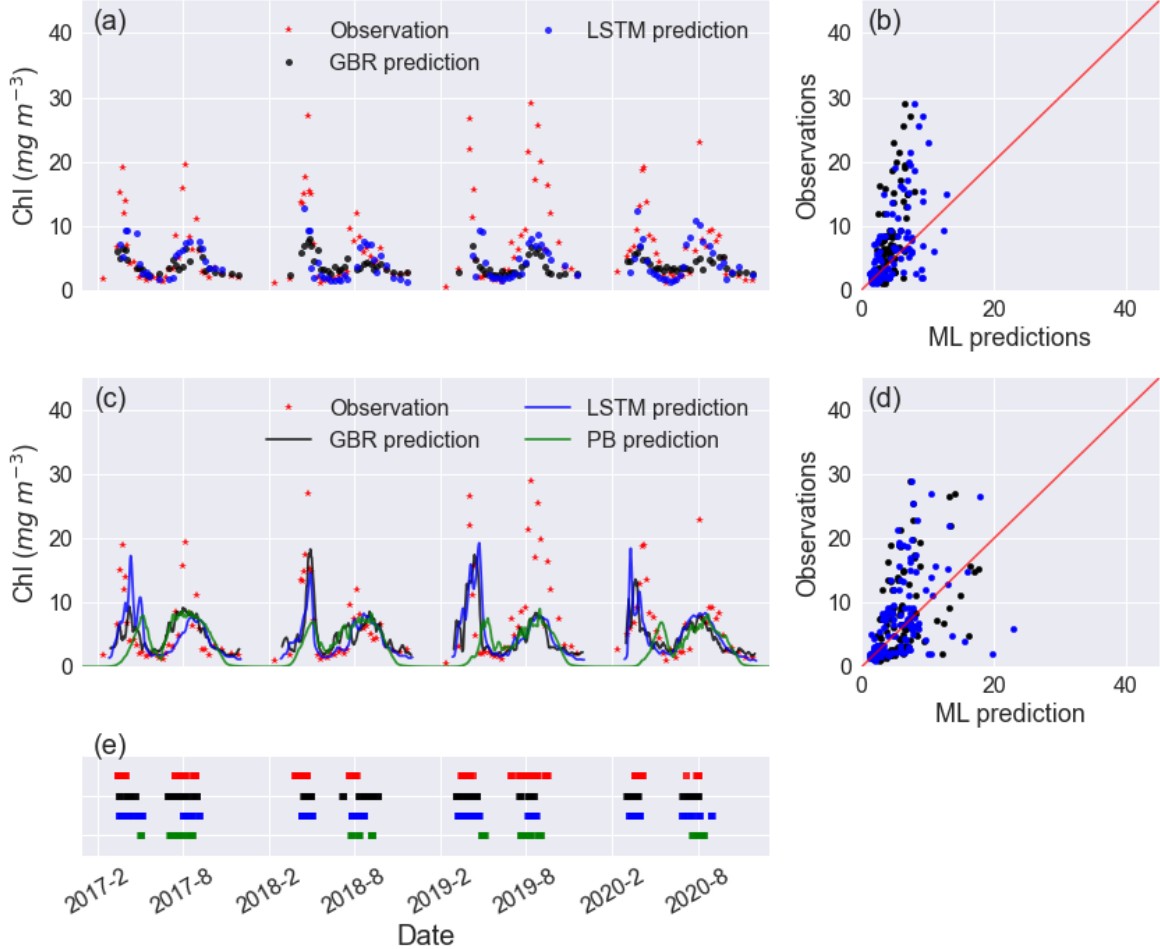


**Figure 2.** Timeseries of observed and predicted *Chl* from GBR and LSTM models in (a) workflow 1 and (c) workflow
3, and the corresponding scatter plots of observations vs ML predictions of *Chl* in workflow 1 and workflow 3 are
shown in panels (b) and (d), with the black and blue dots/lines representing the predictions from GBR and LSTM,
respectively. Panel (e) shows the observed and predicted algal bloom onsets in 2017-2020 using the same color coding
as the previous panels. Results from the PB model simulation in Mesman et al. (2022) are also shown in (c) and (e).

**3.2 Workflow 2: Two-step ML models based on pre-generated daily nutrients and observed physical factors**

As in workflow 1, both ML models in workflow 2 had poor fit in the summer of 2019 and suffered from overfitting
leading to higher *MAE*, *RMSE*, and lower $R^2$ in testing datasets than training datasets (See SI, Table S2).

Overall, both GBR and LSTM showed slightly higher *MAE* (4.22 mg m$^{-3}$ vs. 3.87 mg m$^{-3}$) and *RMSE* (6.27 mg m$^{-3}$
vs. 6.00 mg m$^{-3}$) when compared to workflow 1 (Table 2). But they also showed improved performance in terms of
capturing the peak values of *Chl* during spring blooms (Fig. 2, Fig. S5, SI). Both workflows outperformed the
SELMAPROTBAS PB model in simulating concentrations of lake nutrients (See Fig. S6, SI). The ML models were

more accurate in predicting the low values of $NO_X$ and peak values of $PO_4$ and Total P. However, both ML models
and the PB model failed in predicting the extremely high values of measured lake nutrients, such as the autumn peak
of $NH_4$ in 2017 (Fig. S6e) and the spring peak of $O_2$ in 2018 (Fig. S6c), Thus, higher workflow 2 *MAE* and *RMSE*
(Table 2) are presumably due to the inaccuracies in the pre-generated nutrient training data, but the improved daily
predictions that better capture the bloom events, overshadow these flaws.
**Table 2** Comparisons of model performance during the testing period based on *RMSE*, *MAE*, and *R2*. The unit of *Chl*
is mg m$^{-3}$. In bold are the best fits of each statistical metric. For comparison of training and testing periods, see Table
S2.

| Model | PB | ML-workflow 1 | | ML-workflow 2 | | ML-workflow 3 | |
|---|---|---|---|---|---|---|---|
| | | GBR | LSTM | GBR | LSTM | GBR | LSTM |
| *RMSE* | 7.18 | 5.77 | **5.64** | 6.27 | 6.00 | 5.94 | 5.81 |
| *MAE* | 4.77 | **3.55** | 3.58 | 4.22 | 3.87 | 3.99 | 3.71 |
| *R2* | -0.25 | 0.13 | **0.20** | 0.05 | 0.13 | 0.14 | 0.18 |


**3.3 Workflow 3: based on workflow 2, and including hydrodynamic training features derived from the**
**GOTM model.**
Including hydrodynamic training information in workflow 3 did not significantly improve in lake nutrient predictions
compared to workflow 2 (See Fig. S6), and when using workflow 3 both ML models showed comparable performance
in *Chl* predictions compared to workflow 1. However, the predictions of the spring bloom in all years improved
compared to workflows 1 and 2, in terms of the magnitude and timing of the spring bloom (Fig. 2e). This was the case
in 2019-2020 (Fig. 2a) which was an abnormally warm winter with only 5 days ice cover, and had an unusually early
spring algal bloom. Both GBR and LSTM in workflows 2 and 3 did not capture the extremely intensive bloom (with
peak values close to 30 mg m$^{-3}$) in summer of 2019, and neither did the PB model.
Furthermore, adding hydrodynamic features derived from PB model improved predictions of the onset of algal blooms
(Fig. 2e and 4), with the overall TPR increasing by 15 % and 5 %, FPR increasing around 5% and 3 % in GBR and
LSTM models, respectively. Compared with the PB model which showed lower TPR (15%) and FPR (6%), ML
models are more likely to predict algal bloom at the correct time. The optimal TPR was from LSTM in workflow 3,
which could detect the onset of algal blooms with TPR closed to 50%. However, the concomitant higher FPRs
indicating an incorrect warning of algal bloom is also more likely to occur in the ML models, since the PB model is
more like to miss the bloom entirely. The Kappa values of both ML models and the PB model are close to 80%,
showing that all models simulated the entire period (blooms and the periods between blooms) to a moderate-strong
level (McHugh, 2012).

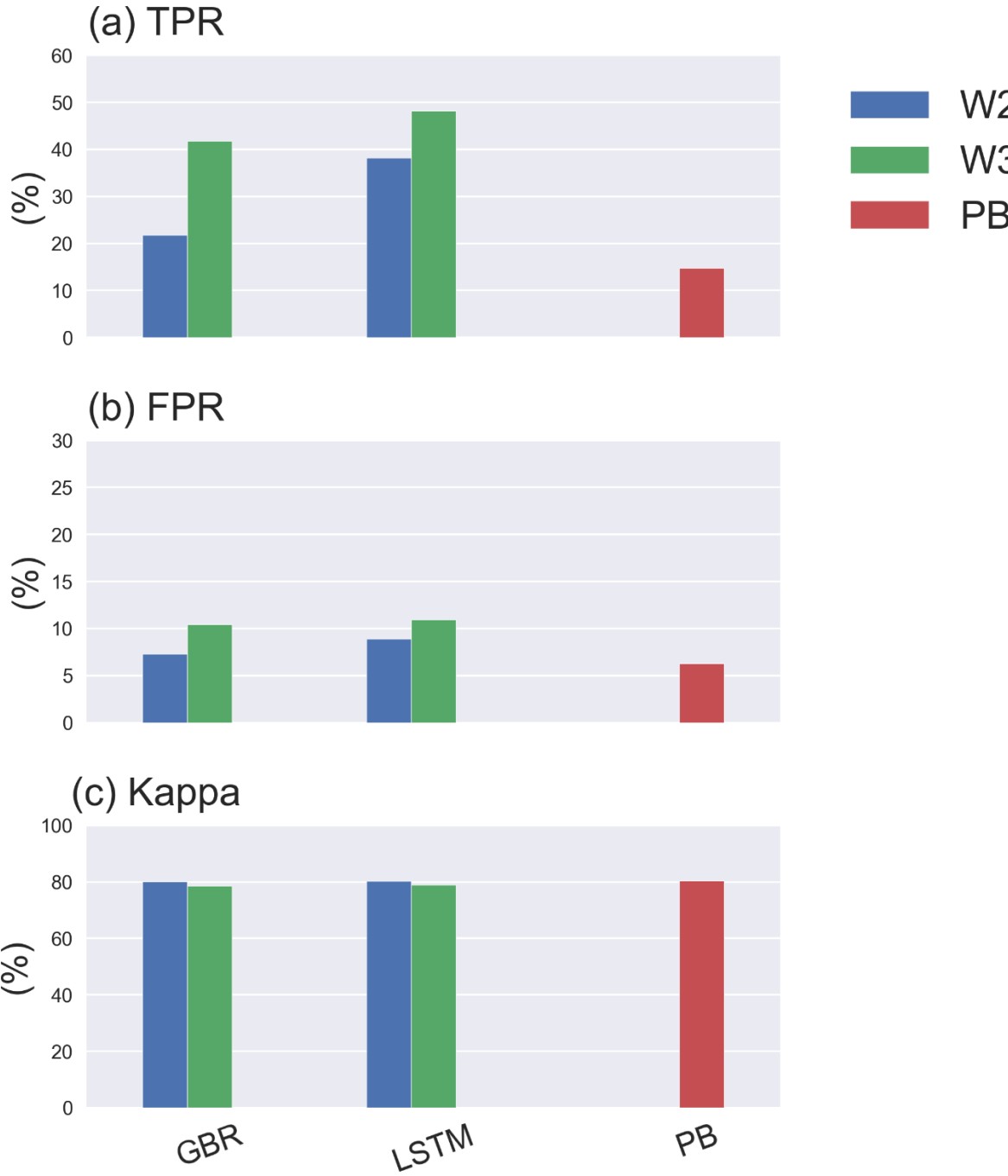


**Figure 3.** TPR, FPR, Kappa of GBR and LSTM models in workflow 2, 3 and the PB model.

**3.4 Effects of shuffling training years on 2019-2020 predictions**

The results presented so far are based on a typical strategy of training ML models for a historical period in this case 2004-2016 and then accessing model performance in a second period between 2017-2020. The accuracies of the model predictions were to some extent related to the range and variability in the training data. To evaluate the importance of this we randomly removed two years from a 2004-2018 training dataset, and made 30 different predictions of *Chl* during 2019-2020 when the models had difficulties predicting spring and summer blooms (Fig 5). When trained with the various shuffled combinations, both ML models were capable of reproducing the seasonal variations in algal *Chl* with a 4.5 % and 5.8 % coefficient of variation (CV) in *MAE*, and a 24.0 % and 16.4 % CV in TPR of GBR and LSTM, respectively (See Table S3, SI). This provides an indication of the uncertainty that may arise as a consequence of differences in the training datasets used for in our workflows. And, it also shows that even a relatively long training period of 13 years can not totally capture the system behaviour in such a way as to lead to nearly similar bloom predictions.

Although none of the model runs captured the intensive summer bloom in 2019, the spring bloom in both years was well represented, especially by LSTM, in terms of timing and magnitude.

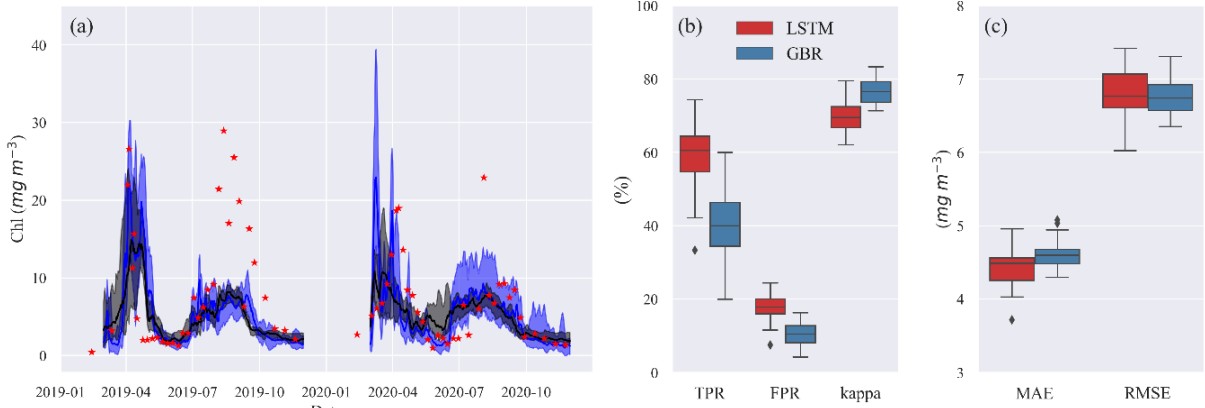

**Figure 4.** (a) Timeseries of observed (red stars) and predicted *Chl* from GBR (black) and LSTM (blue) models in the shuffling training year test. The shades represent the range between minimum and maximum prediction, and the solid lines represent the median prediction. (b) shows the boxplot of TPR, FRP, and Kappa, and (c) shows boxplot of MAE and RMSE of both models in the shuffling training year test.

Despite comparable *RMSE* and *MAE* in LSTM and GBR (Fig. 4c), both higher TPRs (with median of 60%) and FRPs (with median of 18%) in LSTM indicate that the LSTM was more aggressive in making algal bloom predictions. The GBR model's apparent advantage in FPRs (with median 10%) is largely the result of it making a lower number of

bloom predictions since the low concentrations between spring and summer blooms in 2020 was not well represented
(Fig. 4b).

**3.5 Shuffling years data sparsity test**

To examine the possible use of workflow 3 when data are less frequently available, lake nutrient and *Chl* data were
down-sampled so that the effects of sampling frequency on model predictions could be evaluated. Each down-sampled
dataset was also rearranged into 13 different 13-year training periods and 4-year testing periods. The variability in
predictions provided a measure of model performance and uncertainty. Fig. 5 shows the uncertainty in model
predictions as a consequence of the chosen sampling intervals.
The *MAE*s and *RMSE*s of both GBR and LSTM models tended to increase with the longer sample intervals. The
median *MAE* was always slightly higher for the LSTM model except when trained with original dataset (Fig. 5a).
While our initial evaluation of TPR using 2017-2020 as the testing period and 2004-2016 as the training period
suggested the LSTM model was more accurate in turns of detection of algal bloom onsets (Fig. 3), Fig. 5c showed the
median TPR of GBR model calculated by the shuffling year test was over 50%, higher than that found when using the
original testing and training periods. This can be explained by the fact that the 2017-2020 testing  period as in Fig. 3
and shown as large points in Fig. 5 was unusually difficult for GBR to simulate. Consequently, even though the GBR
model usually performs better in the shuffled data test in Fig. 5,  Fig. 3, which shows the results of 2017-2020 testing
period, presented the opposite result. This illustrates the importance of the sequence of training and testing years for
evaluating model performance.
For the first three sampling intervals the GBR model clearly had better TPR values than the LSTM model. The median
TPRs of GBR model started to drop below 30% once the sample interval reached 21 days. For LSTM, medium TPRs
remained lower than 30%, for all sampling intervals but also showed a much wider range of variability (Table S4)
dependent on the training and tested datasets used. In general, both models preformed best at the original and 7-day
sampling interval, but then showed slightly worse performance that was consistent up to a sample interval of 21 days.
In terms of the errors evaluated over the entire 4-year testing period (Fig. 5a, b) the GBR model had lower errors and
therefore, better predicted the seasonal variations of *Chl* concentration. The timeseries comparison of observed and
predicted *Chl* from this shuffling year data sparsity test can be found in SI (Fig. S7-9).

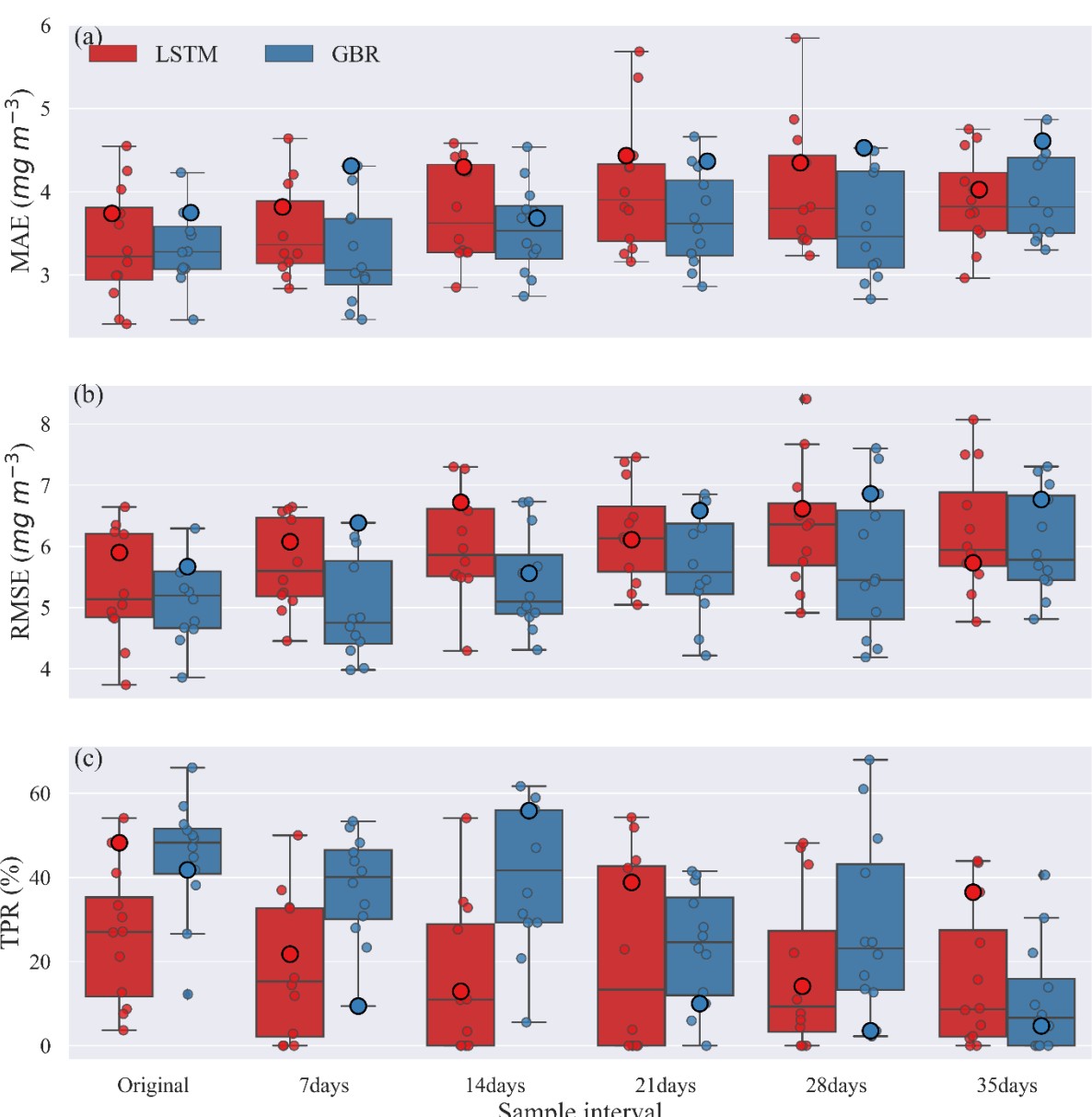

**Figure 5.** Comparisons of (a) *MAE*, (b) *RMSE*, and (c) TPR between GBR and LSTM during the testing period created under various sample intervals. Circles along the box show the result from the testing period of all shuffled training/testing year combinations and the bigger circles represent 2004-2016 training and 2017-2020 testing years combination as was used in Fig. 2.

## 4 Discussion

### 4.1 Performance of ML models

In three workflows, the ML models successfully reproduced the *Chl* seasonal patterns, capturing the spring and summer bloom events, with lower averaged *RMSE*s and *MAE*s than a PB model simulation that was previously

calibrated for Lake Erken. And in all three workflows, LSTM model always showed slightly lower *RMSE*, *MAE* and
higher *R2* in predicting *Chl* concentrations than GBR model, and higher TPR in detecting the onset of algal bloom
events. Workflow 1 which predicted *Chl* based on all available environmental factors including lake nutrient
observations showed that both ML models can reproduce the seasonal dynamics of algal *Chl* with promising accuracy
(*MAE* = 3.55 and 3.58 mg m$^{-3}$, *RMSE* = 5.77 and 5.64 mg m$^{-3}$ and $R^2$ = 0.13 and 0.20, for GBR and LSTM, respectively)
via the direct input of available environmental observations. These ML models can be applied to reconstruct past
patterns of algal *Chl*, fill the gaps between measured *Chl* observations, and interpret the mechanisms that drive
phytoplankton dynamics. Workflows 2 and 3 adopted a two-step approach, first using separate ML models to
estimating daily changes in lake nutrient concentration, and in Workflow 3 also including PB model derived physical
factors as training features of the algal ML model. These two workflows allowed daily predictions of changes in algal
*Chl* concentration using both observations and pre-generated lake nutrient concentrations at a consistent daily time
step, and at only a minor decrease in performance compared to workflow 1, workflow 2 and 3 demonstrated a wider
potential range of applications (e.g., interpolation, reconstruct historical data, algal bloom forecast) via making daily
forecasts with less-than-daily measured nutrient observations.
The one clear failure of both the ML and PB based model predictions was during July-August 2019, *Chl* concentrations
in integrated samples collected between the surface and 6-12 m exceeded 20 mg m$^{-3}$ over a 5-week period. Neither
the PB model nor ML models captured this unusually persistent bloom (Fig. 2, Fig. S3, SI). At this time the
phytoplankton were dominated by the cyanobacteria Gloeotrichia and Anabaena, that form a resting akinete life stage
at the end of their yearly bloom, which can initiate the following year's bloom as they are transformed to vegetative
cells that migrate from the sediment to the upper water column. We hypothesize that the large summer bloom in 2019
was the result of unusually large recruitment of akinetes in this year. (Karlsson-Elfgren et al., 2005; Karlsson-Elfgren
et al., 2004). The life cycle of cyanobacteria is not a process included in the PB model (but see Hense and Beckmann
(2006) and Jöhnk et al. (2011)), so increased recruitment of akinetes could explain the underestimation of the 2019
summer bloom. Even the LSTM algorithms could not account for previous conditions so far back in time as to affect
the formation and deposition of cyanobacteria akinetes (This may require the memory of last ice-free season). The
consequent poor fit of summer bloom in 2019 partially lead to the higher *MAE* and *RMSE* in the testing dataset
compared to the training dataset in all three workflows, in both GBR and LSTM models.
Warm winters can initiate a chain of events, i.e., shortening the ice cover duration, extending spring circulation,
affected nutrients availability, and an earlier spring bloom (Adrian et al., 2006; Yang et al., 2016). According to the
ice record in Lake Erken (See Fig. S1, SI), in 2020, the lake was covered by very thin ice for only 5 days, which is the
shortest duration since observations were first recorded in 1954. The spring bloom in 2020 did occur earlier than other
years (See Fig. S3, SI), and both ML models which considered the timing of lake ice show fairly good performance
in predicting the timing and magnitude of this abnormally early spring bloom (Fig. 2, 5)
4.1.1 Performance of Hybrid PB ML models
One dimensional PB hydrodynamic models can accurately simulate both water temperature profiles, and other
hydrodynamic features in Lake Erken using the same forcing data that are commonly input to ML models. The hybrid
model structure tested here provides a richer set of input data leading to more accurate ML predictions of algal *Chl* at
little additional computational cost or data requirements. Using data from the hydrothermal PB model allowed the
seasonal deepening of the thermocline, variations in the surface mixing layer depth, and upwelling events, represented
by $W_n$, to be encoded into the ML algorithms. These factors can affect the underwater light climate, the internal loading
of phosphorus and the transport of resting cyanobacteria colonies from the hypolimnion into the epilimnion favouring
summer blooms of cyanobacteria (Pierson et al., 1992; Pettersson, 1998). The inclusion of these factors did increase
the accuracy of the ML models, especially in the case of unusual environmental conditions (e.g. spring of 2020, Fig.
2, 5) that did not frequently occur in the remaining meteorological, hydrological and biogeochemical training data.
4.1.2 Prediction of bloom timing
For the purposes of water management, it may be most important to first predict the potential occurrence of a bloom,
and then once underway improve predictions of its magnitude. The best model performance in predicting the timing
of algal blooms, was obtained after adding hydrodynamic features derived from a PB model in workflow 3, with TPR
above 45% in detecting the onset of algal bloom during 2017-2020 and a modified accuracy (Kappa) around 80 %
indicated a moderate – strong level of prediction.
Based on our shuffling year tests of bloom timing, the GBR model showed relatively higher median TPRs than LSTM
model for sample intervals less than one month. However, in some training and testing year combinations, TPRs are
close to 0 % (Fig. 5), and CVs of the TPRs are highly variable, even at the original sample interval, being over 30%
for GBR and over 60% for LSTM, indicating that the correct detection of algal blooms in both models are highly
dependent on the years used to train the models. Thus, while the ML models can be better than the PB models at
predicting the onset of algal blooms, they still may not be good enough for operational forecasting. The resulting
variability provided a more accurate estimate of the model performance at each down-sampled data interval and
showed that increasing sample interval led to reduced performance for both ML models, in terms of *MAE*, *RMSE*, and
the CV of TPR. These tests also highlighted that the performance of both ML models, especially LSTM, varied with
the sampled history of events in the training period for evaluating a specific pattern of change in the testing period.
We suggest that testing strategies similar to the shuffle methods used in this study are needed to accurately evaluate
the expected accuracy of ML models when applied to any given site. The estimated uncertainty in shuffling training
year tests (Fig. 4) and shuffling training/testing year tests (Fig. 5) can be used to better represent the uncertainty of
ML derived forecasts.
**4.2 Future applications in short-term forecasts and water management**
To reach the goal of incorporating ML models into operational forecasts either for short-term management support or
longer-term evaluation and planning, two steps must occur. First the ML model must be developed, trained and
evaluated on the water body of interest due to the unique physical characteristics and water quality dynamics in
different systems. Secondly, future forcing data for the model must be obtained and integrated into a workflow that
makes the future predications. In regards to the second point, a lack of frequent water monitoring (Stanley et al., 2019)
is a major deterrence to applying ML models to many lakes. The data sparsity test (Fig. 5) showed that, at least for
Lake Erken, the ML models can still detect the seasonal algal dynamics even for sample intervals approaching one
month (Fig. S7-9). If this result holds for other lakes, the use of the two-step ML workflow could offer a method of
forecasting seasonal variations in algal *Chl* even in lakes with relatively infrequent nutrient monitoring but higher
frequency meteorological and hydrological data.
The hybrid PB/ML models have the potential to provide reasonably accurate and timely short-term algal bloom
forecasts, working as part of an early-warning systems for the water resource management (Baracchini et al., 2020),
and clearly have the ability to predict border seasonal variations in algal *Chl* concentration. However, since a large
amount of water temperature and water quality samples are required for ML training, and since our results apply to
only one well-studied lake, obtaining more datasets to test and evaluate the workflows developed here are needed.
Monitoring networks (e.g., Global Lake Ecological Observatory Network [GLEON, https://gleon.org/]), could
provide the data to allow more extensive testing and application of hybrid PB/ML models, and we are presently
working in the GLEON network to test the methods developed in this paper on many other lakes.

## 5 Code availability

Model version 1.0 has been archived in Zenodo under DOI:10.5281/zenodo.7149563, and is available at https://github.com/Shuqi-Lin/Erken_Algal_Bloom_Machine_Learning_Model.git.

## 6 Data availability

All data from this study have been archived with the code are also archived in Zenodo under same DOI:10.5281/zenodo.7149563 in the 'training data' folder. Here we also provide the model forcing data in the format used in the machine learning models. Data collected by the Erken laboratory, in the archived format used by the Swedish Infrastructure for Ecosystem Science (SITES) is available from the SITES data archive https://data.fieldsites.se/portal/

## 7 Supplement

## 8 Author contribution

The concept of ML model workflow was designed by SL and DP. SL developed the ML model code and performed the simulations. JM conducted the PB model simulations. SL wrote the manuscript with contributions from DP and JM.

## 9 Competing interests

The contact author has declared that neither they nor their co-authors have any competing interests.

## 10 Acknowledgement

S.L. and this study are funded by the EU and FORMAS project 2018-02771, in the frame of the collaborative international Consortium BLOOWATER (https://www.bloowater.eu/) financed under the ERA-NET WaterWorks2017 Cofounded Call. This ERA-NET is an integral part of the 2018 Joint Activities developed by the Water Challenges for a Changing World Joint Program Initiative (Water JPI). J.P.M. was funded by the European Union's Horizon 2020 Research and Innovation Programme under grant agreements no. 722518 (MANTEL ITN) and 101017861 (SMARTLAGOON). This study has been made possible by the Swedish Infrastructure for Ecosystem Science (SITES), in this case by data from the Erken Laboratory of Uppsala University. SITES receives funding through the Swedish Research Council under the grant no. 2017-00635.

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
