# Peer review of "Prediction of algal blooms via data-driven machine learning models"

_Geoscientific Model Development, 2022_

## Author Response (AR1)

**Reply to CEC:**

The Erken data we used in this study is stored under same DOI:10.5281/zenodo.7149563 as code, in the 'training data' folder. Here we also provide the model forcing data in the format used in the machine learning models. Data collected by the Erken laboratory, in the archived format used by the Swedish Infrastructure for Ecosystem Science (SITES) is available from the SITES data archive https://data.fieldsites.se/portal/

We have included the GPLv3 License within the latest Zenodo repository and fixed the hyperlinks in the manuscript.

**Reply to RC1:**

1. The authors mentioned some machine learning models in the Introduction. These are important in the development of algal bloom prediction. The authors should analyze the disadvantages of these ML models and the improvements of their own model.

   I have added some sentences in Line 37-46 that summarizes the disadvantage of the present ML models, and their limitations in water quality and plankton dynamic prediction. The following paragraph illustrates the improvements we made in our models.

2. The literature review of ML models is too simple, which makes it difficult to find the development of models.

   As I replied to the last comment, I have added the information about the development and application of ML models in water quality and algal bloom prediction (Line 37-46).

3. Page3, Line 73. The significance of designing three workflows needs to be further clarified.

   The significance of designing three workflows has been clarified in the Introduction (Lines 47-53), and the details about the workflow were illustrated in the 2.4 section.

4. Page 3, Line 80. Why do the authors use GBR and LSTM?

   The characteristics and benefits of these two ML has been added (2.3.2; Line 89-105)

5. The advantage of two-step method is accurate prediction when observations are insufficient. However, workflow 1 performs better than workflow 2 or 3 (Table 1). From this comparison, the two-step method is not an important step that affects the accuracy.

   Yes. But workflow 1 can only predict Chl concentration when lake nutrients observations are available, which could be infrequent in most of lakes, and it is also hard to apply this workflow in the algal bloom forecast due to lack of water quality forecast. The advantage of workflows 2 and 3 is therefore a wider potential range of application (e.g., interpolation, reconstruct historical data, algal bloom forecast) at only a minor decrease in performance for this particular lake. The advantages of workflows 2 and 3 were illustrated in the discussion (4.1 Performance of ML models)

6. From Fig. 3 (e.g., Kappa scores), the PB model also works well. What is the advantage of ML models?

The advantages of ML models were revealed by the higher TPR. Although PB works well in terms of Kappa scores, it means PB model can correctly predict chlorophyll concentrations during most of the no-algal bloom period. However, our goal is to capture the algal bloom onset as skillful as we can. ML models have a clear advantage in terms of predicting algal blooms event, which only happened in relatively small proportion of time (Fig. 3a, b).

**Reply to RC2**

**Major remarks**

1. Possible overfitting

The authors discuss the potential overfitting issues a few times. L259, "there was overfitting issues in all three workflows, in both GBR and LSTM models, indicated by higher MAE and RMSE in the testing dataset compared to the training dataset especially for GBR". This statement is not completely accurate: higher error in the testing dataset does not immediately imply overfitting. In particular, the authors discuss the peculiarities of the algal bloom in July-August 2019, which is not properly predicted by any model; as this occurrence is in the testing phase, it means that the errors are expected to be large anyway.

> The overfitting issue means the model is too closely aligned to the training dataset so that it can not predict the peculiarity in the testing dataset. We think this is the issue existing in the three workflows we tested here. The algal dynamics varied from year to year (Fig. S3). If the models show relatively lower RMSE and MAE consistently in the training dataset which include the data from 2004-2016, than testing dataset, overfitting is a likely explanation.

> However, we also agree with your point that peculiarities of the algal bloom in 2019 could also contribute to consistent high errors in testing dataset since the most of observed data points were way higher than usual values. Thus, we adjusted our explanations about the higher RMSE and MAE in testing datasets (Lines 172-175, 189-190, 304-307).

On the other hand, overfitting issues may effectively exist in the model application. In fact, Text S2 reports on the hyperparameters of the LSTM model: by adopting 3 layers and 100 neurons, it approximately implies 300 degrees of freedom (parameters). No information is provided for GBR (please add it).

> We added some information about GBR model in Line 93-97, "The hyperparameters in GBR are optimized via *RandomizedSearchCV* function within Scikit-Learn library. The loss function of model is chosen as '*huber*', which is a combination of the squared error and absolute error of regression. Since the target variable in our research Chl concentration has peak values during algal blooms which could be regarded as outliers, the '*huber*' loss function is more robust and gives greater weight to peak values than the mean squared error function."

How does the high number of parameters to be calibrated in the ML model compare with the number of available data? If the number of data is not large enough, the model is intrinsically prone to overparameterization. Did the authors test different hyperparameters for LSTM, e.g. smaller number of neurons and layers?

> We have tried different combination of numbers of layers and neurons (1-3 layers, 20-200 neurons), but larger numbers of layers and neurons did not obviously improve the results but increased the computational time a lot, and worse results were achieved when the number of layers and neurons were decreased. We added these details into Supporting Information Text S2.

2. Intrinsic variability in the model's results

The authors analyze the variation of the results obtained in the testing period (2019-2020) when shuffling the training years (section 3.4), and of other possible modifications of the dataset, e.g., by artificially reducing the frequency of the data. As I already mentioned, this is very important, and the analysis is well conceived. Nevertheless, single realizations of ML models may provide non-optimal results. For this reason, it is a common practice to repeat ML runs several times and then average the results (e.g., Piotrowski et al., 2021; Yousefi and Toffolon, 2022). Did the authors account for this?

Yes. That's the reason we conducted these two shuffling year tests. As we mentioned in Line 335-338, "We suggest that testing strategies similar to the shuffle methods used in this study are needed to accurately evaluate the expected accuracy of ML models when applied to any given site. The estimated uncertainty in shuffling training year tests (Fig. 4) and shuffling training/testing year tests (Fig. 5) can be used to better represent the uncertainty of ML derived forecasts."

**Minor remarks and typos**

1. The Supporting Information contains some data and plots that would fit well in the main text. For instance, Table S1 is useful to understand the procedure used in the analysis.
   We have moved the table from SI to the main text (Table 1).

2. "Even the LSTM algorithms could not account for previous condition so far back in time". How long is the expected memory of the model?
   L270: The expected memory to consider the formation and deposition of cyanobacteria akinetes may require

   couples of months extending to the previous ice-free season (Lines 304-305).

3. L56 "beings": begins

4. L136-137: "modified Kappa" is not a common metric. Please give a short description of what it represents.

   L136-137: modified accuracy (Kappa) which considers the possibility of the agreement occurring by chance

   (Table S2; McHugh, 2012)

5. L228 "even though the GBR model usually performs better in Fig. 5c the testing period chosen for use in Fig. 3, showed the opposite result." Cumbersome sentence, please rephrase it. Moreover, the whole section contains a weird use of commas.
   L228: Consequently, even though the GBR model usually performs better in most of chosen 4-year testing periods (Fig. 5), Fig. 3, which shows the results of 2017-2020 testing period, presented the opposite result.

6. Figure 4. The subplots (b) and (c) are not described in the caption.
   The caption has been modified.

7. Text S3, reference to Wilson et al. (2020): it is not in the bibliography.
   The bibliography has been updated.

8. Figure S3, specify that boxplots refer to the period 2004-2020. Maybe it would be more interesting to compute the boxplots excluding 2019-2020?
   This was a typo. The boxplots refer to the period 2004-2018.

9. Figure "Penal": Panel:
   Panel (c)

---

## Author Response (AR2)

**Report #1**

The authors proposed a two-step ML method to overcome the limitation of insufficient observational data for algal bloom prediction. There are still three major problems: (1) Unclear certainty of the two-step ML method; (2) The differences between different models applying the two-step ML method were not evaluated; (3) The impact of the predicted data as training data is unknown.

(1) The evaluating metrics (*MAE*, *RMSE*, *R2*) of three workflows were provided in Table 2, indicating the performance of the methods. Regarding the uncertainty, we used the results of the shuffling years test (using different years for training the model) as the main indicator of uncertainty. Two shuffling-year tests (section 2.4) were conducted based on the two-step ML method (workflow 3) to evaluate the uncertainties induced by variations in the data used to train the ML models (section 3.4) and the uncertainties related to sample frequencies of lake nutrient variables (section 3.5). Figures 4 and 5 visualise the uncertainty related to the ML predictions, and Table S3, 4 show the coefficient of variation.

(2) Based on our results, GBR and LSTM generated similar results in the two-step ML method (Fig. 2). LSTM gave slightly better results in predicting *Chl* concentrations in terms of evaluating metrics (Table 2) and was more accurate in predicting algal bloom events in terms of TPR (Fig. 3a). We added a conclusion about the difference between two ML models in sections 3.3 and 4.1 (L207-208, L273-275)

(3) The impact of the predicted data as training data can be seen in the comparison between workflow 1 and workflows 2, 3 (Table 2, Fig. 2), where the usage of real nutrient data is compared with the predicted data. We concluded the impact in section 4.1 (L280-286),

By using predicted nutrient data as training data, workflows 2 and 3 still capture the seasonal pattern of algal *Chl* concentrations. Especially in workflow 3, LSTM predicted the concentration with RMSE =5.81 mg/L, which is very closed to the RMSE in workflow 1 (5.64 mg/L). With only a minor decrease in performance compared to workflow 1, workflows 2 and 3 had a wider potential range of applications (e.g., interpolation, reconstruct historical data, algal bloom forecast) via making daily forecasts with less-than-daily measured nutrient observations.

Also, if you want to know the daily prediction of *Chl* based on physical environmental factors (without any lake nutrient observations), we presented the results from both ML models below. The performance of GBR model under this scenario was much worse than that using the workflows (1-3). In contrast, LSTM model showed fairly good results, comparable to workflow 2.

[Figure]

**Figure.** Timeseries of observed and predicted *Chl* from GBR (a) and LSTM (c) models, and the corresponding scatter plots are shown in panels (b) and (d).

Minor remarks:
1. The authors mentioned the overfitting a few times, whether it's caused by the correlations of the data.

The algal dynamics varied from year to year (Fig. S3). If the models show relatively lower *RMSE* and *MAE* consistently in the training datasets which include the data from 2004-2016, than testing dataset, overfitting is a likely explanation (Table 2).

Thus, we gave mode detailed explanation about the higher *RMSE* and *MAE* in testing datasets in L167-169, 297-299. We only mentioned overfitting twice in the manuscript,

but we have changed L. 181 to "likely suffered from overfitting", as we agree with the reviewer that alternative explanations are possible.

2. In the first step of the two-step ML approach, "each pre-generated nutrient prediction is included in the training data for the next nutrient prediction", the authors should analyze the prediction accuracy of this approach.

We have assessed the accuracy of nutrient prediction by providing the visual time-series comparison and evaluation metrics (Fig. S6). The predictions from ML models for the 6 nutrient variables assessed were better than that from the process-based model (L185-186).

3. The purpose of designing the three workflows is not explained by the authors.

The logic and purpose of designing three workflows has been clarified in the Introduction (Lines 49-56). And the details of designing the three workflows was depicted in section 2.4.

4. Problems with the narrative logic of the literature abstract in this manuscript, the authors mainly used a data-driven model, but this piece is less described.

In the Introduction section (L31-46), we now retraced the applications of ML models in algal bloom studies and mentioned the shortcoming of the ML models, including lack of interpretability and generalization, limitation induced by sparse nutrient monitoring data. And then, we added the main objectives of our study and logic line behind the three workflows in the following paragraph to make the narrative logic clearer (L47-56).

5. The logic of the literature review of ML models is not clear enough, which makes it difficult to find the development of the models.

See the response to the last comment. We have summarized the development of ML models application in algal bloom prediction in the Introduction section (L31-46).

6. Page 2, Line 36-37. Recknagel et al.'s model is based on a neural network, while Nelson et al.'s model is based on a random forest algorithm.

We have corrected the citation here.

7. Subsection 2.3.2. The authors need to conclude the reasons for comparing GBR and LSTM.

We have added a sentence in the last paragraph of this section (L104-106):

'Compared to GBR model, LSTM has more complex model architectures, carrying the 'memory' from the previous time steps. In this study, GBR and LSTM were applied, respectively, to assess the performance of ML models with and without 'memory'.'

**Report #2**

The authors replied to the referees' remarks in an appropriate way.

The only issue that I can highlight is that I could not find a precise correspondence between line numbers in the response and those in the revised manuscript. Also, the reference to Table S2 in the response is not correct (likely it is to the new Table S1), and the reference to McHugh, 2012, is not reported in the text. I invite the authors to be more precise with these details.

We have corrected the original response and the related information in the manuscript. And yes, the reference to Table S2 is to the new Table S1.